# Pathophysiological Mechanisms in Long COVID: A Mixed Method Systematic Review

**DOI:** 10.3390/ijerph21040473

**Published:** 2024-04-12

**Authors:** Nawar Diar Bakerly, Nikki Smith, Julie L. Darbyshire, Joseph Kwon, Emily Bullock, Sareeta Baley, Manoj Sivan, Brendan Delaney

**Affiliations:** 1Faculty of Science and Engineering, Manchester Metropolitan University, Manchester M15 6BH, UK; 2Northern Care Alliance NHS Foundation Trust, Salford M6 8HD, UK; emily.bullock@nca.nhs.uk; 3Locomotion Study Patient Advisory Group, Leeds Institute of Rheumatic and Musculoskeletal Medicine, Level D, Martin Wing, Leeds General Infirmary, Leeds LS1 3EX, UK; nikki.smith@forensic-science.uk.net; 4Nuffield Department of Primary Care Health Sciences, University of Oxford, Oxford OX2 6GG, UK; julie.darbyshire@phc.ox.ac.uk (J.L.D.); joseph.kwon@phc.ox.ac.uk (J.K.); 5Birmingham Community Healthcare NHS Trust, Birmingham B7 4BN, UK; sareeta.baley@nhs.net; 6Rehabilitation Medicine, University of Leeds, Leeds Teaching Hospitals and Leeds Community Healthcare NHS Trusts, Leeds LS11 0DL, UK; m.sivan@leeds.ac.uk; 7Medical Informatics and Decision Making, Imperial College, London SW7 2AZ, UK; brendan.delaney@imperial.ac.uk

**Keywords:** long COVID, post COVID syndrome, pathophysiology, brain fog, fatigue, treatable trait

## Abstract

Introduction: Long COVID (LC) is a global public health crisis affecting more than 70 million people. There is emerging evidence of different pathophysiological mechanisms driving the wide array of symptoms in LC. Understanding the relationships between mechanisms and symptoms helps in guiding clinical management and identifying potential treatment targets. Methods: This was a mixed-methods systematic review with two stages: Stage one (Review 1) included only existing systematic reviews (meta-review) and Stage two (Review 2) was a review of all primary studies. The search strategy involved Medline, Embase, Emcare, and CINAHL databases to identify studies that described symptoms and pathophysiological mechanisms with statistical analysis and/or discussion of plausible causal relationships between mechanisms and symptoms. Only studies that included a control arm for comparison were included. Studies were assessed for quality using the National Heart, Lung, and Blood Institute quality assessment tools. Results: 19 systematic reviews were included in Review 1 and 46 primary studies in Review 2. Overall, the quality of reporting across the studies included in this second review was moderate to poor. The pathophysiological mechanisms with strong evidence were immune system dysregulation, cerebral hypoperfusion, and impaired gas transfer in the lungs. Other mechanisms with moderate to weak evidence were endothelial damage and hypercoagulation, mast cell activation, and auto-immunity to vascular receptors. Conclusions: LC is a complex condition affecting multiple organs with diverse clinical presentations (or traits) underpinned by multiple pathophysiological mechanisms. A ‘treatable trait’ approach may help identify certain groups and target specific interventions. Future research must include understanding the response to intervention based on these mechanism-based traits.

## 1. Introduction

The severe acute respiratory syndrome coronavirus 2 (SARS-CoV-2) has affected the lives of millions of people around the world with devastating consequences in mortality and morbidity. By the end of March 2023, more than 20.5 million people had tested positive in the United Kingdom, with just over 187,000 deaths reported [1].

Most people recover quickly from COVID-19 with symptoms resolving within 2–4 weeks; however, up to 20% continue to suffer from medium to long-term symptoms following the initial infection [2]. In the UK, this number is estimated by the Office for National Statistics to be approximately 1.9 million as of 30 March 2023. Of those, 37% were infected during the Omicron period, 29% had (or suspected they had) COVID-19 in the first months of the pandemic, 17% in the Delta period, and 13% in the Alpha period [3]. Studies suggest that approximately 45% of Long COVID (LC) patients require a reduced work schedule compared to pre-illness and 22.3% are not working due to the illness [4].

The term Long COVID includes, according to the NICE guidelines, symptomatic COVID-19 persistent in 4–12 weeks after infection and post-COVID-19 syndrome, characterized by persisting symptoms over 12 weeks after infection. There are patients who continue to experience LC symptoms for over 3 years [5,6].

Patients with LC have reported more than 60 physical and psychological signs and symptoms, including weakness, general malaise, fatigue, concentration impairment, breathlessness, and patient-reported reduced quality of life [7]. Lung injuries, venous/arterial thrombosis, heart injuries, cardiac/brain stroke, and neurological injuries appear to be the most common ongoing complications following acute COVID-19 infection [8], but presentations are heterogeneous and may occur alone or in various combinations. These varied and diverse clinical presentations of LC often challenge clinicians to devise and implement complex therapeutic plans. Adopting a “treatable trait” approach to LC could both help simplify treatment plans and identify treatment targets for future research.

Several attempts have been made to classify LC presentations into different phenotypes based on symptoms and severity. Some clinical and biochemical characteristics have been defined in patients with acute COVID-19 infection. These have been further classified into various phenotypes, which have helped to inform prognosis and provide tailored clinical management [9,10,11]. Specifically, studies have looked at the biochemical and cellular level to understand the pathophysiology behind LC, such as dysfunction in clotting proteins and the lytic system leading to hyperactivated platelets and circulating micro-clots [12], and persistence of a cytotoxic program evident in CD8+ T cells with elevated production of type 1 cytokines and interleukin-17 (IL-17) [13]. The symptoms severity classification by Sivan et al. between mild, moderate, and severe phenotypes could also help healthcare providers plan services and interventions [14].

Treatment targets continue to lack focus although more recent evidence from acute COVID-19 studies (e.g., PHOSP-COVID [15]) have started to pave the way for therapeutic interventional studies on LC, focusing on well-defined treatable traits (e.g., PHOSP-I looking at the effect of IL-6 inhibitor on inflammatory LC) [16].

Conducting a systematic review in this diverse and fast-moving field is never going to be ‘definitive’, but we believe the time is right for a systematic and transparent synthesis of the literature that attempts to integrate the findings across different domains into a causal network to help identify suitable therapies. We originally intended to conduct a meta-review to inform our research. However, existing systematic reviews were found to be of low quality, and we found recent narrative reviews to be speculative and not grounded in a systematic approach to finding and assessing recent research [17]. Therefore, we conducted a full systematic review of the literature with tight eligibility criteria to include only the most reliable reviews or studies with well-described and relevant control groups.

## 2. Methods

### 2.1. Search Strategy

A search strategy was developed (Appendix B) and was used to search Medline, Emase, Emcare, and CINAHL to identify papers. “Two searches were conducted, an initial search in July 2022 for existing systematic reviews of LC symptoms and pathophysiology (i.e., a meta-review of reviews; henceforth, ‘Review 1’) and a second search in October 2022 for primary studies of LC symptoms and pathophysiology published between June 2021 and October 2022 (‘Review 2’). We used the same search strategy, eligibility criteria, and screening process in Review 2 as in Review 1 with the exception of excluding systematic reviews to capture any additional primary studies that had been published since the coverage period of Review 1, so they could also be included in this paper. To avoid missing any relevant publication, we accepted a degree of overlap in timing between the two searches, considering that the majority of the studies in the systematic reviews included in Review 1 were concluded by June 2021.

Covidence [18] was used to remove duplicates among the imported references. At least two reviewers from a team of seven (BD, JD, NB, EB, NS, JK, CF) independently reviewed each title and abstract and then the full texts on Covidence. At each review stage, if an article received two approvals from any pair of the seven reviewers, it proceeded to the next stage (from title/abstract screening to full-text screening and then to data extraction), with disagreements referred to a third reviewer for the final assessment. The kappa statistics for the inter-rater agreement between all pairs were calculated for each stage, with kappa above 0.400 being considered a moderate-to-good level of agreement.

### 2.2. Eligibility Criteria 

Inclusion criteria included the use of an appropriate definition of LC (e.g., a pre-defined list of LC symptoms warranting clinical diagnosis); participants with a clinically confirmed diagnosis of COVID-19 (e.g., positive PCR test history); studies including patients not hospitalised for acute COVID-19 (i.e., not solely LC patients who were hospitalised); papers including data on symptoms and pathophysiological mechanisms with statistical analysis and/or discussion of plausible causal relationships between mechanisms and symptoms; and appropriate study design with comparator group of general population or non-LC controls, including prospective cohort studies, cross-sectional studies, case-control studies, and randomised controlled studies.

Exclusion criteria included studies including participants with symptom duration of less than 4 weeks following acute COVID-19 infection; unclear COVID-19 diagnosis history, including reliance on antibody test alone for diagnosis; non-English language articles; studies including paediatric participants (<18 years); inappropriate study designs, including lack of a control group, letter to editor and qualitative studies; and conference abstracts and pre-prints (Table 1).

### 2.3. Data Extraction and Synthesis

Data from each included study were extracted independently by any pair from the team of seven reviewers onto the pre-specified extraction template in Covidence. Disagreements from the double extraction were resolved by discussion within the reviewer team. These discussions were scheduled weekly during the study.

The following data were extracted from each systematic review included in Review 1: study ID; title; lead author contact details; study funding sources; possible conflicts of interest; country of study; study aim; start and end dates of the search for systematic review; search strategy for review; study designs included in review; target population; inclusion and exclusion criteria of review; total number of studies included in review; LC symptoms identified from the review; pathophysiological mechanisms identified from the review; and a summary of key discussion points and conclusions reached by review authors.

The following data were extracted from each primary study included in Review 2: study ID; title; study funding sources; possible conflicts of interest; country of study; study aim; start and end dates of study; target population inclusion and exclusion criteria; method of participant recruitment; sample (divided into LC patients and controls) size and characteristics, including the proportion with history of hospitalisation for acute COVID-19 infection; LC symptoms; pathophysiological mechanisms; principle positive and negative associations between symptoms and mechanisms; plausible causal relationships suggested or found by the data analysis; and a summary of key discussion points and conclusions reached by primary study authors.

### 2.4. Quality Assessment

Quality Assessment was completed by one author, with a second author completing a 10% cross-check for validation. Differences in assessment were resolved by group discussion.

Studies that met the definitions for inclusion were assessed for quality based on their design and reporting. In both cases, we considered whether results offered a direct link between symptoms and potential pathophysiology, if data were collected prospectively or retrospectively, and whether the underlying acute COVID infection was self-reported or confirmed by a validated test. We also considered how LC was defined—whether this was self-reported or in line with the World Health Organisation criteria.

For Review 1 (Table 2), we assessed reviews by the population of patients included (removing those which were entirely hospitalised cohorts) and by the extent to which analysis directly linked pathophysiology to symptoms. We excluded reviews of studies that were purely descriptions of symptom patterns. We identified funding sources and possible conflicts of interest declared by the authors. We also considered the study designs included in each review and whether the authors had performed a quality assessment or risk of bias analysis for the publications they included. We did not exclude on any of these latter considerations. Publications included in this paper were assessed for quality using the NIH National Heart, Lung and Blood Institute’s quality assessment tool for Systematic Reviews and Meta-Analyses [19].

For Review 2 (Table 3), we additionally classified studies by the number of participants (<25, 25–100, and >100) and assessed how the population sample was defined and whether the study included a control group and considered potential confounding factors in their analysis. Publications that were included were assessed for quality using the NIH National Heart, Lung, and Blood Institute’s standardised evaluation tool, appropriate for the individual study design [19].

Finally, we used publications from both searches to look at the relationship between symptoms of LC, pathophysiologies underlying these symptoms, and possible treatments or treatment targets. Causal relationships were identified from the extracted data by organ system. 

The reporting of this review was guided by the standards of the Preferred Reporting Items for Systematic Review and Meta-Analysis (PRISMA) Statement.

## 3. Results

### 3.1. Search Results

Figure 1 and Figure 2 present the PRISMA flow diagrams for Reviews 1 and 2, respectively. After screening of titles/abstracts and full texts, 19 systematic reviews were included in Review 1 and 46 primary studies in Review 2. A total of 51 studies for Review 1 and 229 for Review 2 were excluded. The kappa statistics for Review 1 were 0.475 (95% CI 0.345–0.605) for the titles/abstracts screening stage and 0.327 (95% CI 0.094–0.560) for the full texts screening stage. For Review 2, this was 0.393 (95% CI 0.331–0.454) and 0.279 (95% CI 0.150–0.409) for the same two corresponding stages.

### 3.2. Quality Assessment Results

#### 3.2.1. Review 1

By July 2022, 19 systematic review articles were included from our search strategy. One paper focused predominantly on case studies of patients with hair loss [27]. All other papers included a broad spectrum of study design, most (including the paper by Hussain et al. in 2022) [27] set no limits on design within their search criteria. Not all reviews were clear whether studies included hospitalised and non-hospitalised cases of COVID-19; however, where this was highlighted, most reviews included patients from both groups of patients. Of the 19 reviews included in this first review, most (n = 13) declared no funding source. The remaining six were supported by academic or national funding bodies [7,21,23,34,36,37]. Two publications declared specific potential conflicts of interest, including affiliations with pharmaceutical companies leading to payment for speaking, consultation, or research activities [33,35]. Eight of the nineteen publications reported that titles and abstracts were reviewed by just one author [25,26,28,29,30,32,35,36].

Nine of the reviews included in Review 1 considered publication bias [20,21,23,25,28,29,31,35,36]. Overall, the quality of reporting across the reviews was poor. Five did not have a clear and focused question [21,25,29,30,36], and seven did not adequately define and specify inclusion/exclusion eligibility criteria [25,28,29,30,31,35,36]. Four studies did not report a systematic approach [23,25,28,36], and only eight reviews included a comprehensive table listing the characteristics of patients who participated in the included studies [7,24,27,28,32,33,34,37].

See Table 2 for a quality assessment of all nineteen papers included in Review 1, and Appendix A in the online supplement for a summary of the findings from this review.

#### 3.2.2. Review 2

Forty-six studies with a total of 10,921 participants were included in the second review. Studies included were published between June 2021 and October 2022 (17 from the USA and 21 from Europe). See Table 3 for a quality assessment of all publications included in Review 2, and Appendix A in the online supplement for a summary of the findings from this review. Of these, 19 and 15 were case-controlled and cohort studies with control groups, respectively. All studies included patients not hospitalised during their acute COVID-19 event, with 21 studies clearly including patients hospitalised for acute COVID-19. Four studies recruited less than 25 patients [38,49,77,78], whilst another four included more than 1000 participants [44,47,60,75].

Overall, the quality of reporting across the studies included in this second review was moderate to poor. In particular, only four studies reported a sample size justification [39,44,56,79] and only two studies randomised the control group [60,76].

Most studies (n = 38) used a control group that was at least similar to the LC population, but nine studies did not clearly differentiate their control and intervention groups [45,46,52,57,59,63,67,70,75]. Most studies were able to determine that presenting symptoms developed after acute COVID (i.e., were not existing symptoms prior to infection), but just eleven considered potential confounding variables in their analysis [47,54,58,61,63,65,66,67,72,74,80]. One study did not present a clear research question or objective [65], with only seven studies analysed without the assessors knowing to which study group each participant belonged [45,46,55,56,57,63,69].

#### 3.2.3. Symptoms Strongly Linked to Identifiable Pathophysiological Pathway Immune System

The link between the SARS-CoV-2 infection and immune dysregulation has been established in a number of studies in the acute phase [84]. This relationship appears to be part of the development of LC; albeit, the relationship between LC and the immune system is complex.

Symptoms of fatigue, postural orthostatic tachycardia syndrome (POTS), and neurocognitive dysfunction have all been linked to immune dysregulation. Some studies [48,65,67,74,76,80,85] and systematic reviews [20,22,23,24,25,26,27,30,31,35] suggested a relationship between Interleukins (ILs) 6 and 8, IP 10, and TNF alpha with LC symptoms of fatigue and neurocognitive dysfunction.

In one study, participants with LC symptoms of fatigue, brain fog, memory difficulty, confusion, and POTS also displayed altered humoral responses to distinct herpes viruses, including the Epstein-Barr Virus (EBV) minor viral capsid antigen gp23, the EBV fusion receptor component gp42, and the VZV glycoprotein E [61].

Viral persistence with reduced viral cleavage leading up to higher levels of monocytes [71] or a maintained T cell response to SARS-CoV-2 in LC patients has been reported in more recent papers [49], whilst a systematic review by Akbarialiabad in 2021 suggested that oxidative stress leading to weak immune response and incomplete viral eradication is associated with LC symptoms, including neural loss associated with the finding of SARS-CoV-2 RNA in the CNS [20]. In a small study of 49 LC patients mainly suffering from systemic symptoms, chest pain, or fatigue compared with 16 controls, the level of the CD4 count was found to be lower in the LC population [55]. Similar findings implicating the CD8 pathway were noted in a case-controlled study of 30 LC patients and 20 controlled with a significant increase in the levels of CD8+ T cells for the LC group relative to the recovered group [53].

Another study by Lee and colleagues associated symptoms of fatigue and dyspnoea with the involvement of interferon responsive genes in the pathophysiology of COVID-19, indicating a possible link to systemic autoimmune diseases [63]. Additionally, one systematic review, including studies of hospitalised and non-hospitalised patients with COVID-19 infection, identified complex pathways to LC. These pathways included Micro and Macro vascular inflammation associated with increased levels of cytokines, circulating endothelial cells, coagulation activation, and microvascular retinal impairment. Some studies included in this review found biopsy evidence of lymphocytic or neutrophilic infiltrates, endotheliitis, microangiopathy, and microthrombosis [23].

Finally, one study using a metabolic phenotyping approach on stored sera from non-hospitalised LC patients compared healthy controls and sera from hospitalised acute COVID-19 patients. Results were indicative of ongoing inflammation, cellular damage, and immune activity in the LC patients (raised taurine, 3-indole acetic acid, glutamine/glutamate ratios) [59].

##### Central Nervous System

Neurocognitive symptoms are one of the most commonly reported findings in LC. Three systematic reviews [29,33,36] and four recent papers [40,43,68,81] reported links between neurocognitive symptoms and brain pathophysiology. Whilst there is evidence of brain microbleeds and structural damage in patients dying of severe COVID-19, evidence in the non-hospitalised population takes the form of positron emission tomographic (PET) imaging. One high-quality-controlled PET-MRI study included in a review of the relationships in LC sufferers between pain (including headache) [36] and general cognitive function [29] found persistent focal cerebral hypometabolism (frontal, temporal, brainstem, and cerebellum) in this group. Two additional studies show similar findings, although a large proportion of patients in these studies had been hospitalised [78,79]. A case-control study of 26 patients with new onset neurocognitive and mood symptoms and 125 healthy retrospective controls used PET-MRI. Hypometabolism of the fronto-insular cortex, along with hypermetabolism of the limbic system, correlated with both markers of inflammation in the acute phase and the Mini-Mental State Examination [68].

Small preliminary studies have also suggested changes in cerebral blood flow, possibly due to endothelial dysfunction. Other studies included in the review found that small white matter lesions on imaging were associated with delayed recovery from mild COVID-19 [36]. A study of cognitive impairment and high-resolution MRI in mildly affected patients with COVID-19 compared with healthy volunteers found significant cortical thinning in areas of language and difficulties with verbal memory [43]. Further studies of brain post-mortem samples in patients who died showed that SARS-CoV-2 was capable of infecting astrocytes and that infected astrocytes impacted neuronal viability in vitro. Further evidence of structural changes in the brain comes from a case-control study comparing high-resolution MRI and neuropsychiatric instrument scores in 30 LC subjects with neuropsychiatric symptoms and 20 healthy volunteer controls [40]. The study found significant associations between increased grey matter volume and depressive and cognitive symptoms. A review of depression symptoms after COVID included two prospective cohort studies [33] that found associations between depressive symptoms and poorer performance on selective attention and processing speed, immediate recall, visual reaction times, executive function, and visuospatial abilities. The direction of causality in these studies cannot be determined.

Linking neurocognitive LC symptoms with abnormalities in the immune response, a case-control study showed wide-ranging differences in the immune response compared with healthy controls [81]. Broad activation of T helper cells to SARS-CoV-2 nucleocapsid proteins and impaired CD8 T cell memory were found and also found to be associated with the severity of cognitive impairment.

##### Respiratory System

Pulmonary involvement due to the SARS-CoV-2 infection, with the resulting respiratory failure and acute respiratory distress syndrome (ARDS), is considered to be the leading cause of death from acute COVID-19 [86]. Therefore, lung damage, as a consequence akin to SARS-1 infection, was a concern early in the pandemic [87].

In this review, LC patients with persisting dyspnoea, cough, and chest pain following COVID-19 infection were found in a case-controlled study to have significantly lower forced vital capacity (FVC%), total lung capacity (TLC%), and diffusing capacity of the lungs (DLCO%) compared to controls [64]. Other studies also demonstrated similar findings [69,70,77].

Impaired gas transfer was also demonstrated by Xenon MRI scanning in LC patients many months after the acute infection in both hospitalised and non-hospitalised patients despite normal CT imaging of the lungs [57]. A number of systematic reviews also described this association between symptoms of LC (mainly breathlessness and cough) with lung damage demonstrated by either imaging or impaired lung function [22,34,37]. Although lung damage leading to impaired gas transfer has been demonstrated in these studies and systematic reviews as a pathophysiology underpinning symptoms of dyspnoea, cough, or fatigue, there is still little evidence describing the causality of this lung damage, albeit micro-clots and endothelial damage have been proposed by some [69].

##### Other Direct Organ System Damage

Our review identified two systematic reviews [32,37] and three new primary studies [42,44,75] linking non-respiratory organ damage to disease pathophysiology.

A systematic review of LC effects on multiple systems found a number of imaging studies in LC patients, although most were uncontrolled cohorts, and the majority were probably in hospitalised patients [37]. This review pointed to myocarditis and peri-myocarditis being the primary areas of concern outside the lungs. A high-quality systematic review specifically looking at this issue found 35 studies (22 solely in hospitalised patients) reporting on a variety of cardiac assessments, including MRI, echocardiography, NT ProBNP, ECG endomyocardial biopsy, and angiography [32]. Cardiac MRI (cMRI) evidence of myocarditis was found in 0–37% of patients in studies, whereas ECG and echocardiography were usually normal. It was unclear from the review what the impact of hospitalisation was on pathophysiology. Four controlled individual studies have examined the impact of LC on the heart [42,56,62,75,83], whilst another study focused on the heart and other organs [44]. Mostly cMRI with perfusion and gadolinium has been used. A study of military personnel with LC symptoms compared to healthy controls found that 4/50 of the cases met the criteria for myocarditis at 1–4 months post-infection [42]. Follow-up scans showed resolution in ¾ of the cases by 9 months. One case of Takotsubo cardiomyopathy was found (ischaemic damage without any evidence of coronary artery disease). A prospective cohort study of volunteers at the point of COVID-19 infection found reduced exercise capacity on cardiopulmonary exercise testing in those developing LC symptoms [42]. The study found this reduced capacity was associated with increased perceived effort and chronotropic incompetence. There was no evidence of abnormalities in cMRI (stress-perfusion MRI) or rhythm monitoring; reduced exercise capacity was associated with raised CRP, IL-6, and TNF alpha, linking back to a non-cardiac cause. Two smaller studies [56,62] also found no significant evidence in cMRI. A much larger study involving a whole-body MRI in 536 individuals with LC and 92 healthy controls found evidence that mild structural organ damage was relatively common (myocarditis 9% pancreas 9% kidney 15% liver 11%) and 2/3rds had at least one organ inflamed at the initial scan, although these changes were not associated with symptomology statistically [44].

#### 3.2.4. Symptoms with Reasonable Link to Identifiable Pathophysiological Pathway

##### Endothelial and Blood Clotting Disturbances

Three systematic reviews [23,28,35] and three studies [41,60,78] have established a relationship between endothelial dysfunction and LC. One additional high-quality study has shown abnormalities in blood clotting mechanisms [52]. A systematic review of LC identified a number of studies where macro- and microvascular thrombosis played a part in pulmonary and cardiovascular complications, but it is unclear to what extent these occurred in hospitalised patients alone [28]. A review of erectile dysfunction after COVID-19 noted that endothelial function is necessary for erection, but the papers within this review were not quality scored nor were they explicitly looking at erectile function [35].

In a more recent study, PET was added to whole-body imaging in a case-controlled study of a small sample of 13 patients in an LC clinic compared with non-contemporaneous controls (cancer patients). A mild/moderate increased uptake was seen in vessels, bone marrow, and joints in LC patients, but no correlations were found with symptoms, though this analysis would likely lack the power [78]. The effects of SARS-CoV-2 on the vascular endothelium (measured by flow mediated skin fluorescence, FMSF) and exercise capacity have been investigated in 49 patients from an LC clinic who were compared with healthy controls and a group of amateur runners [41]. FSMF measures nitric oxide mediated vessel dilation in response to reactive hyperaemia and small vessel oscillations. High-intensity exercise and LC have similar effects on blood vessel function, except that the effect of high-intensity exercise resolved after several hours. Although no direct correlations with symptoms were made, the researchers noted that the vascular changes after exercise are known to be associated with post-exercise fatigue. In the only RCT in our review, volunteer patients with at least 4 weeks of symptoms post-COVID were randomised to multivitamins or L-arginine and a high dose of vitamin C, proposed as being able to reduce oxidative stress and improve endothelial function. There were significant improvements after 30 days in a range of LC-related symptoms and in Borg Scale perceived effort [60]. Micro-thrombi in the pulmonary vasculature is well recognised in patients who have died of acute COVID-19. A case-control study in patients attending an LC clinic was conducted to determine the role of endothelial damage using biomarkers for endothelial cell damage (von Willebrand Factor Antigen (VWF:Wg), VWF propeptide (VWFpp), and soluble thrombomodulin (sTM)). The study included 13/50 patients who had not been hospitalised. Levels of all these markers and Factor VIIIC were found to be significantly raised in LC patients. Abnormalities were associated with, but not restricted to, hospital admission in the acute phase. An attempted correlation with symptoms was underpowered.

##### Mast Cells

Mast cells may become activated as a side-effect of T cell activation and many LC patients report new allergic responses, such as wheeze, gastrointestinal symptoms, and urticaria. There is no accepted international definition of mast cell activation, with some authors proposing a symptom-based definition and others restricting the term to the identification of raised serum tryptophan (itself transient and not widely available as an assay). One paper using the clinical definition found that LC patients had similar Mast Cell Mediator Release Scores to clinically confirmed mast cell patients [82].

#### 3.2.5. Symptoms with Weak Evidence Linking Them to Identifiable Pathophysiological Pathway

Several small studies provide speculative and non-significant data in the areas of gene expression relating to anti-viral responses in the immune system [63], differences in the microbiome [54,59], and possible auto-immunity to vascular receptors as a cause of the dysautonomia observed in a number of LC subjects [21,23].

### 3.3. Summary

LC is a complex condition affecting multiple organs with diverse clinical presentations (or traits) underpinned by multiple pathophysiologies. Our findings suggest that significant numbers of people with LC symptoms demonstrate identifiable causes and that the symptoms of LC are potentially driven by a complex causal network. Examination of the biomarkers and pathophysiological mechanisms identified in the papers leads to some suggestions as to future avenues for research in both patient stratification and potential therapies (Table 4).

## 4. Discussion

Using external evidence, including known biological relationships, of potential causal links involving the pathologies identified by this review, symptoms, and identifiable physiological traits in LC patients, we created the causal network shown in Figure 3. This proposed network emphasises the need to capture a wide variety of symptoms and to explore relationships between potential treatments. One implication is that single therapies directed at one aspect of the network may not be effective as other pathways may exist; consider, for example, fatigue, which has many potential causes. In particular, we have placed dysautonomia at the centre of the causal map, as it is likely that for at least a proportion of patients, this represents a significant treatable trait [88].

One of the challenges in linking pathophysiologies to treatment targets is the lack of certainty about the causal effect of a proposed pathophysiology. For example, cognitive impairment with associated CNS inflammation could be due to this inflammation or to another pathophysiology causing both abnormalities. However, we believe that sufficient evidence exists for immunological pathways with persisting inflammation, depletion of CD4 and NK cells, and persisting populations of SARS-CoV-2 specific CD8, leading to immune dysregulation with raised IL6 and TNF Alpha. This may be triggered by viral persistence in the gut and monocytes, but other mechanisms have been proposed. Direct organ damage to the lung, myocardium, pancreas, and kidney may possibly be mediated via endothelial dysfunction or direct effects of the virus, leading to pulmonary fibrosis, peri-myocarditis, diabetes, and hypothyroidism. Neurocognitive problems arise from CNS inflammation with CSF oligoclonal bands and damage to astrocytes or cerebral metabolic dysfunction linked to endothelial dysfunction. Clearly, individual symptoms can also interact with relationships between disturbed sleep, fatigue, executive function, and occupational and social impairment. We propose that mental health problems arise in response to impaired functioning, but mental health-related symptoms may also arise from CNS inflammation. Less good evidence currently exists for mast cell activation as a pathophysiology (though symptomatic urticaria and other histamine-related symptoms can be treated) and the role of activated platelets and clotting abnormalities. For these areas, better-controlled studies are required. Furthermore, remitting/relapsing LC has been described [4], but the pathophysiology remains poorly understood. Our review did not find sufficient evidence to help better characterise this phenotype of LC; although, it appears some relapses were associated with physical activity or stress in the pattern of ‘post-exertional symptom exacerbation’ that is well-described in myalgic encephalomyelitis/chronic fatigue syndrome (ME/CFS).

The strength of our mixed methods systematic review is that it only included studies with well-described links between symptoms and underlying pathophysiologies. We excluded papers with an assumptive nature, and all papers included had to have a control group. We also used an adaptive method to capture all up-to-date evidence by using a two-phase approach to include more recently published studies.

Previous systematic reviews also failed to distinguish clearly between hospitalised and non-hospitalised populations. Disentangling LC from ‘post ICU syndrome’ with its associated deconditioning and potential for interstitial lung disease is challenging, so we only included primary research papers that reported case mix and included non-hospitalised individuals. We note the low kappa scoring in our meta-analysis, which is likely to be due to the complexity and limited specificity of our eligibility criteria (especially an analysis linking pathophysiology and symptoms which we ended up interpreting broadly) and the diverse background of the research team, including non-clinicians. That said, it is possible we may have missed relevant pathophysiologies when we excluded some studies either because they have only included hospitalised patients or did not include controls.

The causal aspect of our review is based on a range of non-systematic review articles, but since the mechanisms of interest have a large literature base prior to LC (dysautonomia, immune regulation, thrombosis, imaging etc.), a fully integrative approach to all domains would have been impracticable. As such, we may have missed findings in original research prior to June 2021 that were not highlighted by the 19 systematic reviews, especially as these reviews were quite broad in scope. Conversely, the 19 existing systematic reviews took a much less stringent approach to inclusion criteria, many including a wide range of case reports and uncontrolled studies. We were careful, therefore, to ensure that our findings were well-supported by the 46 recent high-quality original studies from the 2nd part of our review. We did not include several studies that have received a lot of attention on social media given the small numbers of individuals recruited to many of these or due to the lack of a case-control design. We accepted non-contemporaneous healthy controls or where control samples were collected in patients presenting with a condition that could not in any way be plausibly connected to a pathway in LC. This excluded studies with controls who had another long-term condition. Ideally, studies would have contemporaneous healthy controls without LC symptoms, accepting that most of the population has now had COVID-19 but also should not have had COVID-19 recently in case of asymptomatic abnormalities being present. Another significant difficulty lies in the diagnosis of anxiety and depression. These conditions were usually diagnosed on the basis of the above threshold scores in psychometric instruments, such as HADS and PHQ9. These reference intervals are based on healthy populations, not conditions where symptoms may conflate answers in the instrument. We, therefore, discounted papers where a diagnosis was not made separately by a mental health professional. Most studies recruited from LC clinics and used some form of available controls and only occasionally contemporaneous. There was little attempt in most studies to recruit population-representative samples and reporting of social and ethnic diversity is poor. Finally; long-term data remain lacking to date. Most of the studies and reviews included or followed up with patients for up to 12 months and the longer-term impact remains unclear, although in one questionnaire-based Scottish study (CISS), LC symptoms were found to persist for up to 18 months following the acute infection [89].

## 5. Conclusions

Interventional treatment trials in LC remain lacking. Our review has highlighted potential therapeutic targets, such as immune activation, endothelial dysfunction, and clotting abnormalities, with therapies, including biological agents or the use of anticoagulation as potential interventions, for future research.

Another gap we have uncovered is the lack of evidence on persistent Long COVID and those suffering with it beyond 18 months. We found no significant study of its prevalence, patients’ characteristics, or symptoms that tend to persist the most. These patients continue to suffer at a time when there is reduced political and public attention to it despite the continued impact on the health service, workforce, and the economy.

LC remains a major burden on the health of those affected by it and on healthcare systems. Such a condition affects many systems with diverse clinical presentations. It is extremely likely that LC is underpinned by multiple pathophysiologies, and the key to successfully treating LC is to unlock these pathophysiologies, accepting that some symptoms (or traits) could be caused by multiple pathophysiologies which may co-exist in one patient.

As such, a “treatable trait” approach needs to be considered for therapeutic interventions. Our review is the first in-depth review, to our knowledge, in linking LC symptoms to underlying pathophysiologies in robustly conducted clinical trials, including non-hospitalised, as well as hospitalised patients.

## Figures and Tables

**Figure 1 ijerph-21-00473-f001:**
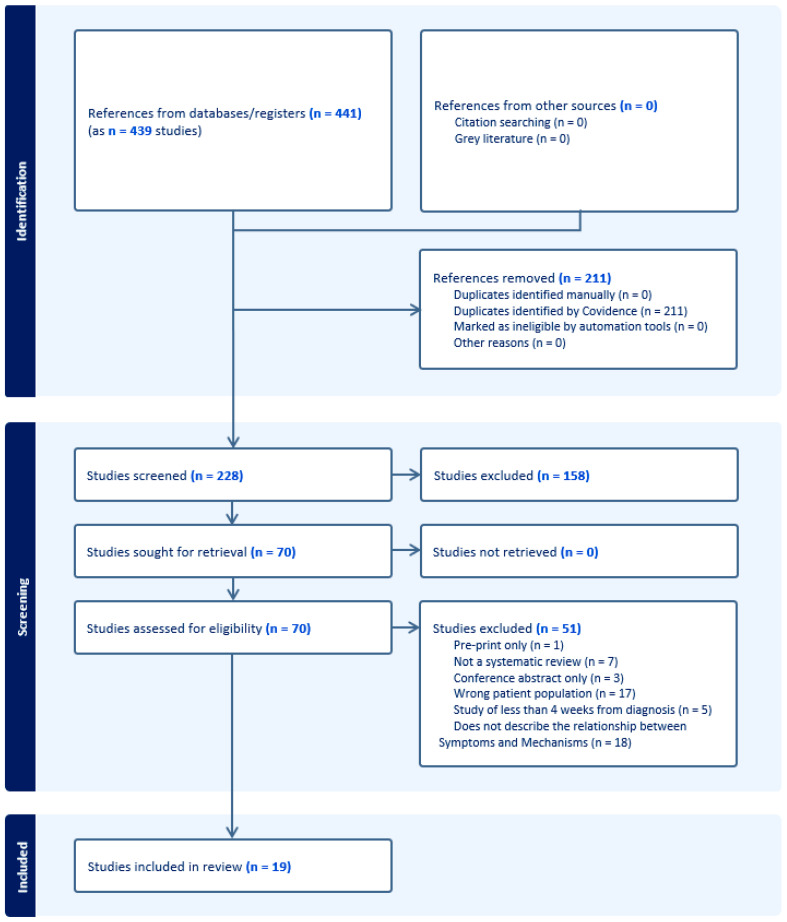
The PRISMA flow diagram, Review 1.

**Figure 2 ijerph-21-00473-f002:**
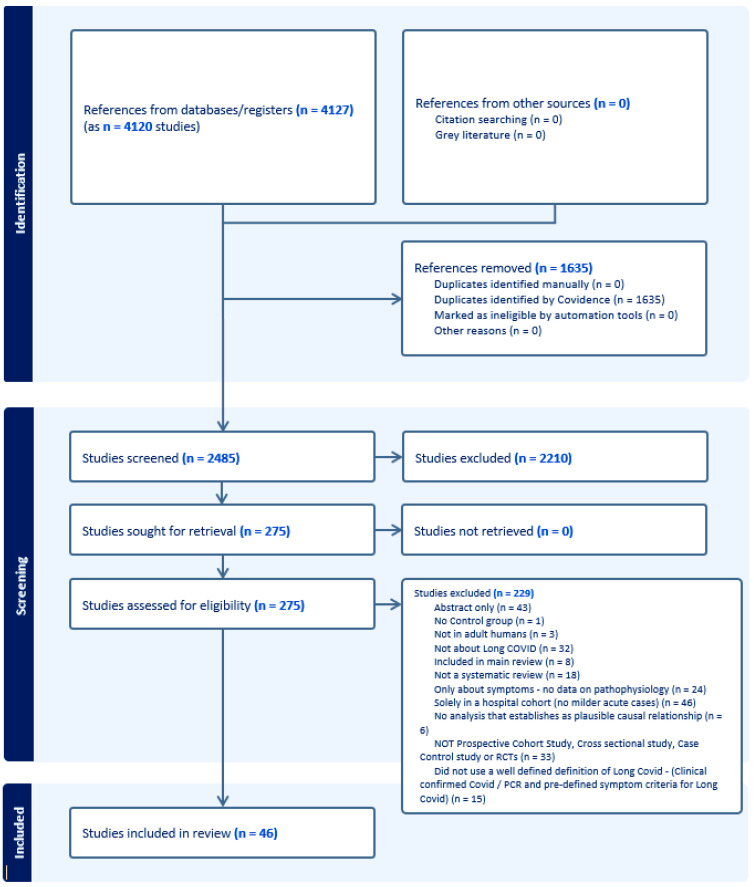
The PRISMA flow diagram, Review 2.

**Figure 3 ijerph-21-00473-f003:**
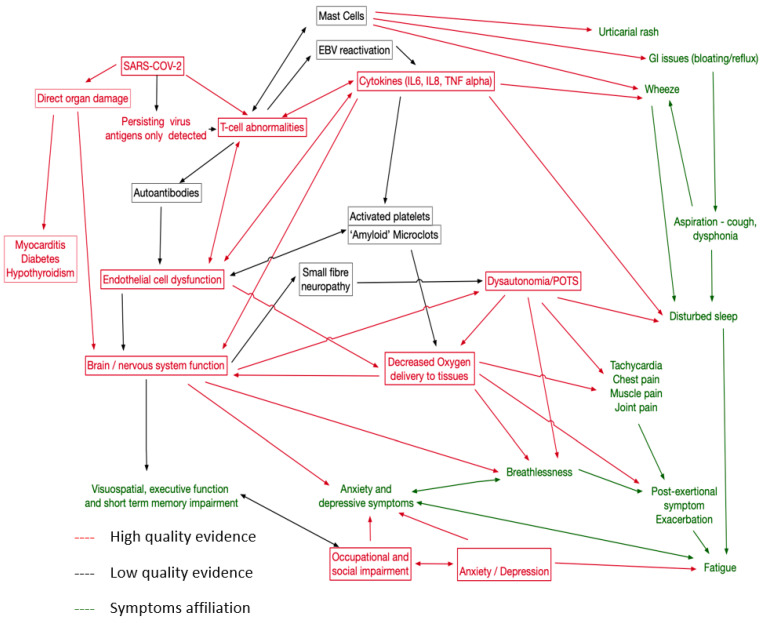
Proposal causal network map for LC. Red for mechanisms well supported in this review and elsewhere, black for less strong evidence. Symptoms in green.

**Table 1 ijerph-21-00473-t001:** Inclusion and exclusion criteria.

Inclusion	Exclusion
Confirmed diagnosis of C19 Hospital and community samples Pre-prints (2020 onwards)—subsequently excluded papers which not subsequently published	None COVID-19 data Unknown time points or less than 4 weeks of recorded symptoms Non-English language Paediatric sample Letter to editor, opinion pieces, conference abstract and posters (if unable to locate paper) Narrow/low generalisability, i.e., small sample size (<100)

**Table 2 ijerph-21-00473-t002:** Quality assessment for Review 1.

Study ID	Title	Is the Review Based on a Focused Question That Is Adequately Formulated and Described?	Were Eligibility Criteria for Included and Excluded Studies Predefined and Specified?	Did the Literature Search Strategy Use a Comprehensive, Systematic Approach?	Were Titles, Abstracts, and Full-text Articles Dually and Independently Reviewed for Inclusion and Exclusion to Minimize Bias?	Was the Quality of Each Included Study Rated Independently by Two or More Reviewers Using a Standard Method to Appraise Its Internal Validity?	Were the Included Studies Listed along with Important Characteristics and Results of Each Study?	Was Publication Bias Assessed?
Akbarialiabad 2021 [20]	Long COVID, a comprehensive systematic scoping review.	Yes	Yes	Yes	Yes	No	No	No
Anaya 2021 [21]	Post-COVID syndrome. A case series and comprehensive review	No	Yes	Yes	Yes	No	No	No
Bergantini 2022 [22]	Common Molecular Pathways Between Post-COVID19 Syndrome and Lung Fibrosis: A Scoping Review.	Yes	Yes	Yes	Yes	Yes	No	No
Castanares-Zapatero 2022 [23]	Pathophysiology and mechanism of long COVID: a comprehensive review.	Yes	Yes	No	Yes	No	No	No
Ceban 2022 [24]	Fatigue and cognitive impairment in Post-COVID-19 Syndrome: A systematic review and meta-analysis.	Yes	Yes	Yes	Yes	Yes	Yes	No
Garg 2021 [25]	The conundrum of ‘long-covid-19: A narrative review	No	No	No	No	No	No	No
Houben 2022 [26]	The Impact of COVID-19 Infection on Cognitive Function and the Implication for Rehabilitation: A Systematic Review and Meta-Analysis	Yes	Yes	Yes	No	Yes	Yes	No
Hussain 2022 [27]	A systematic review of acute telogen effluvium, a harrowing post-COVID-19 manifestation.	Yes	Yes	Yes	Yes	Yes	Yes	No
Joshee 2022 [28]	Long-Term Effects of COVID-19	Yes	No	No	No	No	Yes	No
Meyer 2022 [29]	Molecular imaging findings on acute and long-term effects of COVID-19 on the brain: A systematic review.	No	No	Yes	No	No	No	No
Michelen 2021 [7]	Characterising long COVID: a living systematic review.	Yes	Yes	Yes	Yes	Yes	Yes	Yes
Pierce 2022 [30]	Post-COVID-19 Syndrome.	No	No	Yes	No	Yes	No	No
Piri 2021 [31]	A systematic review on the recurrence of SARS-CoV-2 virus: frequency, risk factors, and possible explanations.	Yes	No	Yes	No	No	No	No
Ramadan 2021 [32]	Cardiac sequelae after coronavirus disease 2019 recovery: a systematic review.	Yes	Yes	Yes	Yes	Yes	Yes	No
Renaud-Charest 2021 [33]	Onset and frequency of depression in post-COVID-19 syndrome: A systematic review.	Yes	Yes	Yes	Yes	Yes	Yes	No
Salamanna 2021 [34]	Post-COVID-19 Syndrome: The Persistent Symptoms at the Post-viral Stage of the Disease. A Systematic Review of the Current Data	Yes	Yes	Yes	Yes	Yes	Yes	No
Sansone 2022 [35]	The Sexual Long COVID (SLC): Erectile Dysfunction as a Biomarker of Systemic Complications for COVID-19 Long Haulers.	Yes	No	Yes	No	No	No	No
Tesarz 2022 [36]	Pain, the brain, and SARS-CoV-2: Evidence for pain-specific alterations in brain-related structure-function properties	No	No	No	No	No	No	No
Willi 2021 [37]	COVID-19 sequelae in adults aged less than 50 years: A systematic review.	Yes	Yes	Yes	Yes	Yes	Yes	No

**Table 3 ijerph-21-00473-t003:** Quality assessment for Review 2.

Study ID	Number of Participants	Was the Research Q or Objective in This Paper Clearly Stated and Appropriate?	Was the Study Population Clearly Specified and Defined?	Did the Authors Include a Sample Size Justification?	Were Controls Selected or Recruited from the Same or Similar Population That Gave Rise to the Cases (Including the Same Timeframe)?	Were the Definitions, Inclusion and Exclusion Criteria, Algorithms or Processes Used to Identify or Select Cases and Controls Valid, Reliable, and Implemented Consistently Across All Study Participants?	Were the Cases Clearly Defined and Differentiated from Controls?	If Less than 100 Percent of Eligible Cases and/or Controls Were Selected for the Study, Were the Cases and/or Controls Randomly Selected from Those Eligible?	Was There Use of Concurrent Controls?	Were the Investigators Able to Confirm That the Exposure/Risk Occurred Prior to the Development of the Condition or Event That Defined a Participant as a Case?	Were the Measures of Exposure/Risk Clearly Defined, Valid, Reliable, and Implemented Consistently (Including the Same Time Period) across all Study Participatns?	Were the Assessors of Exposure/Risk Blinded to the Case or Control Status of Participants?	Were Key Potential Confounding Variables Measured and Adjusted Statistically in the Analyses? If Matching Was Used, Did the Investigators Account for Matching during Study Analysis?
Apple 2022 [38]	22	Yes	Yes	No	Yes	Yes	Yes	No	No	No	Yes	No	No
Beaudry 2022 [39]	66	Yes	Yes	Yes	Yes	Yes	Yes	No	No	Yes	Yes	No	No
Besteher 2022 [40]	50	Yes	Yes	No	Yes	Yes	Yes	No	No	Yes	Yes	No	No
Chudzik 2022 [41]	103	Yes	Yes	No	Yes	Yes	Yes	No	No	No	Yes	No	No
Clark 2021 [42]	100	Yes	Yes	No	No	Yes	Yes	No	No	No	No	No	No
Crunfli 2022 [43]	26	Yes	No	No	Yes	Yes	Yes	No	No	No	Yes	No	No
Dennis 2023 [44]	2460	Yes	Yes	Yes	No	Yes	Yes	No	No	No	No	No	No
Durstenfeld 2022 [45]	102	Yes	Yes	No	Yes	Yes	No	No	No	No	Yes	Yes	No
Durstenfeld 2023 [46]	60	Yes	Yes	No	Yes	Yes	No	No	No	No	Yes	Yes	No
Fancourt 2022 [47]	1457	Yes	Yes	No	Yes	Yes	Yes	No	Yes	No	Yes	No	Yes
Ferrando 2022 [48]	60	Yes	Yes	No	Yes	Yes	Yes	No	Yes	No	Yes	No	No
Files 2021 [49]	24	Yes	Yes	No	Yes	Yes	Yes	No	Yes	Yes	Yes	No	No
Flaskamp 2022 [50]	44	Yes	Yes	No	Yes	Yes	Yes	No	Yes	Yes	Yes	No	No
Finlay 2022 [51]	50	Yes	No	No	No	No	Yes	No	No	Yes	Yes	No	No
Fogarty 2021 [52]	67	Yes	No	No	Yes	Yes	No	No	No	No	Yes	No	No
Galan 2022 [53]	50	Yes	Yes	No	Yes	Yes	Yes	No	Yes	Yes	Yes	No	No
Giron 2022 [54]	217	Yes	Yes	No	Yes	Yes	Yes	No	No	Yes	Yes	No	Yes
Glynne 2022 [55]	65	Yes	Yes	No	Yes	Yes	Yes	No	No	Yes	Yes	Yes	No
Gorecka 2022 [56]	30	Yes	Yes	Yes	Yes	Yes	Yes	No	No	Yes	Yes	Yes	No
Grist 2022 [57]	38	Yes	No	No	Yes	Yes	No	No	No	Yes	Yes	Yes	No
Guo 2022 [58]	421	Yes	Yes	No	Yes	Yes	Yes	No	Yes	No	Yes	No	Yes
Holmes 2021 [59]	86	Yes	No	No	No	No	No	No	No	No	No	No	No
Izzo 2022 [60]	1390	Yes	Yes	No	Yes	Yes	Yes	Yes	No	Yes	Yes	No	No
Klein 2022 [61]	215	Yes	No	No	Yes	Yes	Yes	No	No	Yes	Yes	No	Yes
Kravchenko 2021 [62]	83	Yes	Yes	No	Yes	Yes	Yes	No	No	Yes	Yes	Yes	No
Lee 2022 [63]	182	Yes	Yes	No	Yes	Yes	No	No	No	Yes	No	No	Yes
Lehmann 2022 [64]	135	Yes	Yes	No	Yes	Yes	Yes	No	No	Yes	Yes	No	No
Littlefield 2022 [65]	60	No	Yes	No	Yes	Yes	Yes	No	No	Yes	Yes	No	Yes
Maamar 2022 [66]	121	Yes	Yes	No	Yes	Yes	Yes	No	No	Yes	Yes	No	Yes
Maes 2022 [67]	125	Yes	No	No	Yes	Yes	No	No	No	Yes	Yes	No	Yes
Martini 2022 [68]	26	Yes	Yes	No	No	Yes	Yes	N/A	No	Yes	No	No	No
Matheson 2022 [69]	34	Yes	Yes	No	Yes	Yes	Yes	No	No	Yes	Yes	Yes	No
Munker 2022 [70]	76	Yes	Yes	No	Yes	Yes	No	No	No	Yes	Yes	No	No
Patterson 2021 [71]	144	Yes	No	No	Yes	Yes	Yes	No	No	Yes	Yes	No	No
Peluso 2021 [72]	121	Yes	Yes	No	Yes	Yes	Yes	No	No	Yes	Yes	No	Yes
Peluso 2022 [73]	121	Yes	Yes	No	Yes	Yes	Yes	No	No	Yes	Yes	No	No
Peluso 2022 [74]	280	Yes	Yes	No	Yes	Yes	Yes	No	No	Yes	Yes	No	Yes
Roca-Fernandez 2022 [75]	1151	Yes	No	No	Yes	Yes	No	No	No	Yes	Yes	No	No
Schultheiss 2021 [76]	318	Yes	Yes	No	Yes	Yes	Yes	Yes	No	Yes	Yes	No	No
Singh 2021 [77]	20	Yes	Yes	No	No	Yes	Yes	No	No	Yes	No	No	No
Sollini 2020 [78]	20	Yes	No	No	No	Yes	Yes	No	No	No	No	No	No
Sollini 2021 [79]	39	Yes	Yes	Yes	No	Yes	Yes	No	No	Yes	Yes	No	No
Talla 2022 [80]	101	Yes	Yes	No	Yes	Yes	Yes	No	No	Yes	Yes	No	Yes
Visvabharathy 2021 [81]	159	Yes	Yes	No	Yes	Yes	Yes	No	No	Yes	Yes	No	No
Weinstock 2021 [82]	352	Yes	Yes	No	Yes	Yes	Yes	No	No	Yes	Yes	No	No
Yu 2022 [83]	50	Yes	Yes	No	Yes	Yes	Yes	No	Yes	Yes	Yes	No	No

**Table 4 ijerph-21-00473-t004:** Identified Long COVID pathophysiologies and potential treatment targets.

Pathophysiology Identified	Biomarkers	Potential Treatment Target	Suggestions for Trials
Persisting virus	Viral RNA, Viral proteins	Live SARS-CoV-2	Antivirals
Immune activation	Flow cytometry and phenotyping panels, antigen specific activation studies, IL6, IL8 TNF Alpha	TNF Alpha	Anti TNF Alpha class drugs
Autoantibodies	ELISA panels	Specific autoantibodies	Specific auto-antibody therapy (BC007-aptamer)
Endothelial damage	Reactive hyperaemia and brachial artery ultrasound. CRP	endothelial function and inflammation	Statins, Colchicine, risk prevention, PD-E5 inhibitors, rehabilitation.
CNS damage	FDG-PET	Treat inflammation and endothelial damage	As a cause
Activated platelets	Platelet activation tests. P-selectin	Anti platelet therapy	Anti platelet therapy
Clotting abnormalities	Staining and microscopy or cell counting for microclots, VWF:Ag ratio	Factor Xa	DOAC
Mast cell activity	Clinical scoring	H1 receptors	Antihistamines
Dysautonomia	Lean test for POTS	Cardiac rate limitation, volume expansion	Non-pharmacological, Beta blockers, Ivabradine, Midodrine, Fludrocortisone, rehabilitation
Decreased oxygen delivery	Mixed venous oxygen	Tissue oxygenation	Hyperbaric Oxygen Therapy
Lung damage	DLCO, Xe MRI	Treat associated pathology	As cause.

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
