# Peer review of "Pathophysiological Mechanisms in Long COVID: A Mixed Method Systematic Review"

_ijerph, 2024, doi:10.3390/ijerph21040473_

Round 1

Reviewer 1 Report

Comments and Suggestions for Authors

The article is a forwarded manuscript. The authors have improved some aspects of the manuscript, but I think it still has important flaws:

The authors make two reviews. They create the first review from previous systematic reviews and the second from original articles. However, this methodology is strange as a new systematic review cannot assume that previous reviews contain all the evidence available up to 2021. Authors may consider previous reviews to have been complete and exhaustive. However, what they say in the introduction is that existing systematic reviews were found to be of low quality.

On the other hand, they have really done a single review because they have used the same search strategy and inclusion and exclusion criteria for both reviews, which is questionable.

I disagree with the authors that measuring the quality of articles with a single reviewer with a 10% cross-check from the second author for validation is adequate. The authors have specified the instrument they used to measure quality in review one but not in review 2. Did they use the same one?

In one of their responses, the authors respond that narrative reviews are also systematic, something I'm afraid I have to disagree with. Including systematic and non-systematic reviews in the review of evidence without differentiating between them does not seem appropriate.

The authors say the standards of the PRISMA statement have guided them. If so, they should add a PRISMA Checklist among the supplementary files.

Author Response

Reviewer #1

“The authors make two reviews. They create the first review from previous systematic reviews and the second from original articles. However, this methodology is strange as a new systematic review cannot assume that previous reviews contain all the evidence available up to 2021. Authors may consider previous reviews to have been complete and exhaustive. However, what they say in the introduction is that existing systematic reviews were found to be of low quality.”

We thank the reviewer for their comment. We have already made it clear in the text that existing reviews included in “Review 1” were of low quality as the studies included were small, often uncontrolled and from a relatively narrow time slot at the start of the recognition of Long Covid as a clinical entity. As such, perhaps this was not surprising.

The reason we conducted “Review 2” was primarily a time update. In which case using the same search strategy (but unfiltered by review) was acceptable and scientifically plausible in our view.

“On the other hand, they have really done a single review because they have used the same search strategy and inclusion and exclusion criteria for both reviews, which is questionable.”

We thank the reviewer for this note. We would like, however, to clarify that this is not the case as the 2 reviews were different as they have used different “pool” of publications; and were defined by different timelines as already clarified in the paper under our aims.

I disagree with the authors that measuring the quality of articles with a single reviewer with a 10% cross-check from the second author for validation is adequate. The authors have specified the instrument they used to measure quality in review one but not in review 2. Did they use the same one?

We thank the reviewer for their comments. We would like to point out that we have referenced the evaluation tools in both places in the methods section (ref 18). We have also added details in both places to make it very clear that we used the same battery of tools for both reviews, and that the NIH battery includes quality evaluation guides for different study designs. All changes are tracked in the updated manuscript.

In one of their responses, the authors respond that narrative reviews are also systematic, something I'm afraid I have to disagree with. Including systematic and non-systematic reviews in the review of evidence without differentiating between them does not seem appropriate.

We thank the reviewer for their comments which we also agree with. We did not include ’narrative reviews’ in “Review 1”; instead, we included ’narrative synthesis’ of systematic reviews, the appropriate method here for a very diverse literature that cannot yet be subject to numerical synthesis. 

The authors say the standards of the PRISMA statement have guided them. If so, they should add a PRISMA Checklist among the supplementary files.

We thank the reviewer for their comments. We have used the following standards for our quality assessment. https://www.nhlbi.nih.gov/health-topics/study-quality-assessment-tools

This is already referenced in the manuscript as Ref no. 18.

Reviewer 2 Report

Comments and Suggestions for Authors

I accept the revised manuscript.

Author Response

Dear Ms Chen

We hope this finds you well.

Further to our previous submission to your journal, we are delighted to learn that reviewer 2 has no further comments to add and accepted the changes we have made to our manuscript taking their previous comments into account.

Thanks again for considering our paper for publication.

Prof. Nawar D Bakerly on behalf of the authors.